# Impact of Covid-19 pandemic on obstetric fistula repair program in Zimbabwe

**Chipo Chimamise**[1]*, **Stephen Peter Munjanja**[2], **Mazvita Machinga**[1], **Iris Shiripinda**[1]

**1** Department of Health Sciences, College of Health, Agriculture and Natural Sciences, Africa University, Mutare, Zimbabwe, **2** Department of Obstetrics and Gyynaecology, College of Health Sciences, University of Zimbabwe, Harare, Zimbabwe

☯ These authors contributed equally to this work.
* chipochimamise@gmail.com

**Data Availability Statement:** All relevant data are within the paper and its Supporting information files.

**Funding:** The authors received no specific funding for this work.

## Abstract

The advent of Covid-19 pandemic adversely affected many programs worldwide, public health, including programming for obstetric fistula were not spared. Obstetric fistula is an abnormal connection between the vagina and the bladder or the rectum resulting from obstetric causes, mainly prolonged obstructed labour. Zimbabwe has two obstetric fistula repair centers. Because the program uses specialist surgeons from outside the country, the repairs are organized in quarterly camps with a target to repair 90 women per quarter. This study aimed at assessing the impact of restrictions on movement and gathering of people brought about by the Cocid-19 pandemic and to characterize participants of the camp which was held in the midst of the Covid-19 pandemic at Mashoko Hospital. Specifically it looked at how Covid-19 pandemic affected programming for obstetric fistula repair and characterized participants of the fistula camp held in November to December 2020 at one of the repair centers. A review of the dataset and surgical log sheets for the camp and national obstetric fistula dataset was conducted. Variables of interest were extracted onto an excel spreadsheet and analyzed for frequencies and proportions. Data were presented in charts, tables and narratives. The study noted that Covid-19 pandemic negatively affected performance of fistula repairs greatly with only 25 women repaired in 2020 as compared to 313 in 2019. Ninety women were called to come for repairs but 52 did not manage to attend due to reasons related to the restriction of the Covid-19 pandemic lockdown. Two thirds of those women suffered from urinary incontinence while the other third had fecal incontinence. The successful repair rate was 92%. This study concluded that the pandemic greatly affected programming of fistula repair in the country and recommended the Ministry of Health and Child Care to institute measures to resume programming as soon as the situation allows.

## Introduction

Obstetric fistula, an abnormal connection between the vagina and either the bladder (vesicovaginal fistula) or the rectum (rectovaginal fistula) presents untold suffering to affected women [1]. The condition results mostly from ischemic trauma caused by prolonged and obstructed

**Competing interests:** The authors declare that no competing interests exist.

labour [2]. It affects all aspects of the woman's life including physical life, spiritual, emotional, social and economic life [3]. Worldwide, obstetric fistula affects between 500,00 and 100,000 women every year, with Africa and Asia bearing the highest burden [4].

In Zimbabwe, the actual burden of obstetric fistula is not known. The country started an obstetric fistula repair program as a public health intervention in August 2015 at one provincial hospital. Before then, obstetric fistula repair was mainly available at private hospitals and it was out of reach of many financially compromised women. As efforts to strengthen the fistula program, the Ministry of Health and Child Care piloted a community fistula surveillance system in Manicaland province which is being driven by community based workers.

The program has since been expanded to one more hospital, Mashoko Christian Hospital in Masvingo province and since then more than 700 obstetric fistula survivors have been repaired. Mashoko hospital has up to now conducted five camps. Since the program mostly makes use of visiting surgeons from other countries, fistula repair activities are organized periodically. Patients are identified and put on a waiting list. Fistula repair dates are scheduled usually on a quarterly basis and patients are called for repair on the scheduled dates, this is termed a fistula repair camp.

Zimbabwe registered its first Covid-19 case on the 21st of March 2020 and the country went into a lockdown with stringent regulations which restricted movement and gatherings. This affected the fistula camp which was organized to start that week where a total of 90 women were expected to be repaired at the two fistula repair sites. The lockdown regulations were relaxed in July to allow limited gatherings and more free movement of people. But Chinhoyi Provincial Hospital was not open for fistula camps mainly because staff and resources had been drained due to the Covid-19 pandemic and part of the hospital was designated as a Covid-19 wing, so there was no space to hold a fistula camp. A camp was called for at Mashoko hospital and was scheduled for November to December 2020 even though there was limited freedom on movement of people and other infection prevention and control measures.

To strengthen the fistula program, it is crucial to have information on beneficiaries of the program so that interventions are tailored to the needs of the beneficiaries, such as their ages, when they develop the fistula and whether the program is effective in bringing relief from their suffering.

The aim of this study was to document effects of the Covid-19 pandemic on the fistula repair program in Zimbabwe and to characterize participants of the obstetric fistula repair camp held in November to December 2020 at Mashoko Christian Hospital.

Its objectives were to determine the extent to which Covid-19 pandemic affected fistula repair programing, to determine demographic characteristics of the camp's participants in relation to where they came from, who referred them for the camp, their problems and surgical outcomes.

## Methods

### Study design and study setting

A retrospective cross sectional study was conducted using analysis of program data on the obstetric fistula repair camp conducted at Mashoko Christian Hospital and national obstetric fistula database.

### Study datasets

The dataset contains information and data elements on: name of patient, village, district and province of origin, age, period staying with fistula, who referred the patient, whether it is a new or old patient, number of previous fistula repairs, type of injury and type of fistula,

surgical diagnosis and surgical procedure performed, surgical outcome and number of days in hospital, among others. These were the data elements needed for this study. Personal identifying data elements such as name, hospital number, contact details and national identity number, were removed from the dataset before analysis to maintain confidentiality. The national fistula program dataset was also analyzed to assess the trends in fistula repairs per quarter in the year compared to the same period the previous year.

### Sampling and sample size

Records of all the thirty patients who attended the November-December 2020 fistula repair camp at Mashoko Hospital were purposively included in the study. The national fistula repair dataset was also analyzed for camp attendance for 2019 and 2020.

### Data collection

An excel spreadsheet with data elements of interest was prepared and data were extracted from the surgical log sheet onto the spreadsheet. Completeness of the data was assured during the process of data extraction by cross checking the log sheet against the patient clinical records. Key informant interviews were also held with the hospital superintendents of the fistula repair hospitals and the National Obstetric Fistula Officer to get information on program implementation in the context of Covid-19.

### Data analysis and presentation

Since the volume of the data was relatively small, data were analyzed using the excel spreadsheet for frequencies, proportions and odds ratios. Information from the key informant interviews were analyzed manually for content. The data were presented in tables, figures and narratives.

### Ethical considerations

This article is part of a lager study approved by the Medical Research Council of Zimbabwe under license MRCZ A2525. This study mainly used a records review approach, reviewing the history of how the patients were identified and other demographic data from their medical records excluding their personal identifications.

Written informed consent was obtained from all study participants. Key informants gave written informed consent. Confidentiality was maintained throughout the study.

## Results

The Covid-19 pandemic hindered the holding of fistula repair camps in 2020. There was only one camp held with only 25 fistula cases and 5 perineal tears repaired. Fig 1 below illustrates the comparison of fistula cases repaired in 2020 when Covid-19 resulted in restrictions on movement and gathering of people against 2019 when such restrictions were not there.

A total of 90 women with urine and/or fecal incontinence were called for the repair camp, 82 confirmed that they were coming. Of these 52 did not reach the facility because of various reasons related to Covid-19 lockdown restrictions. Fig 2 below shows the numbers of women called for the camp, those who could not come for various reasons and those who actually came.

Reasons given for not reporting for the camp included:

- Could not get travel authorization documents—16

- Afraid of catching Covid-19–8

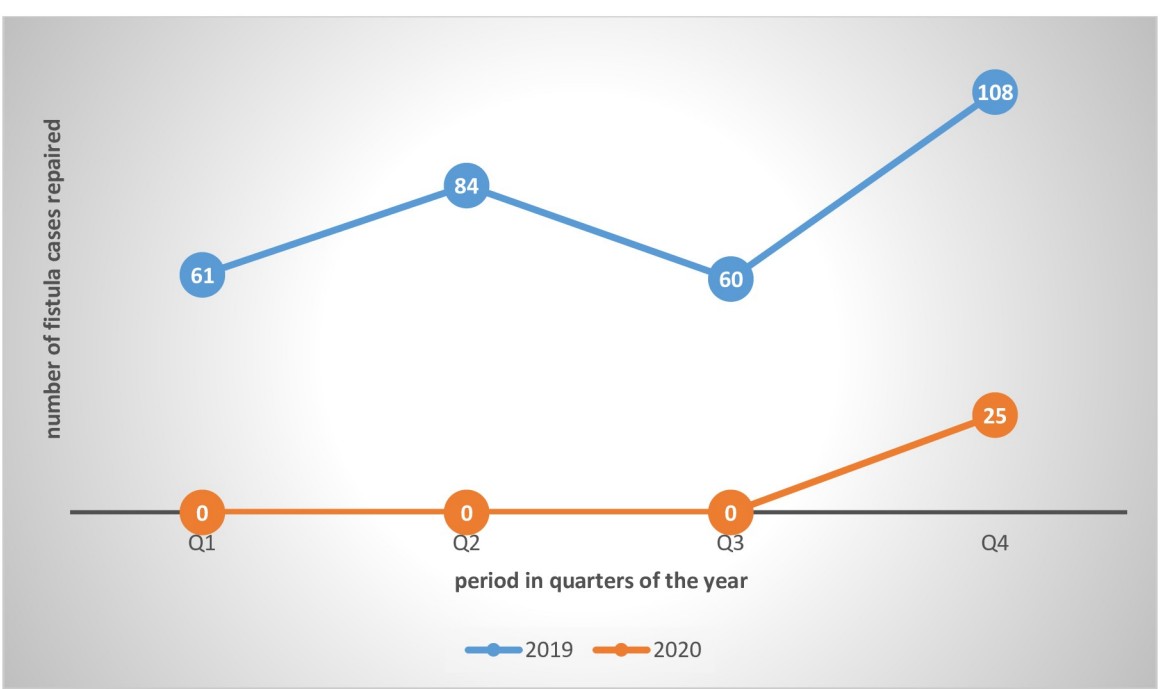

**Fig 1. Trends in fistula camp attendance before and during the Covid-19 pandemic.**

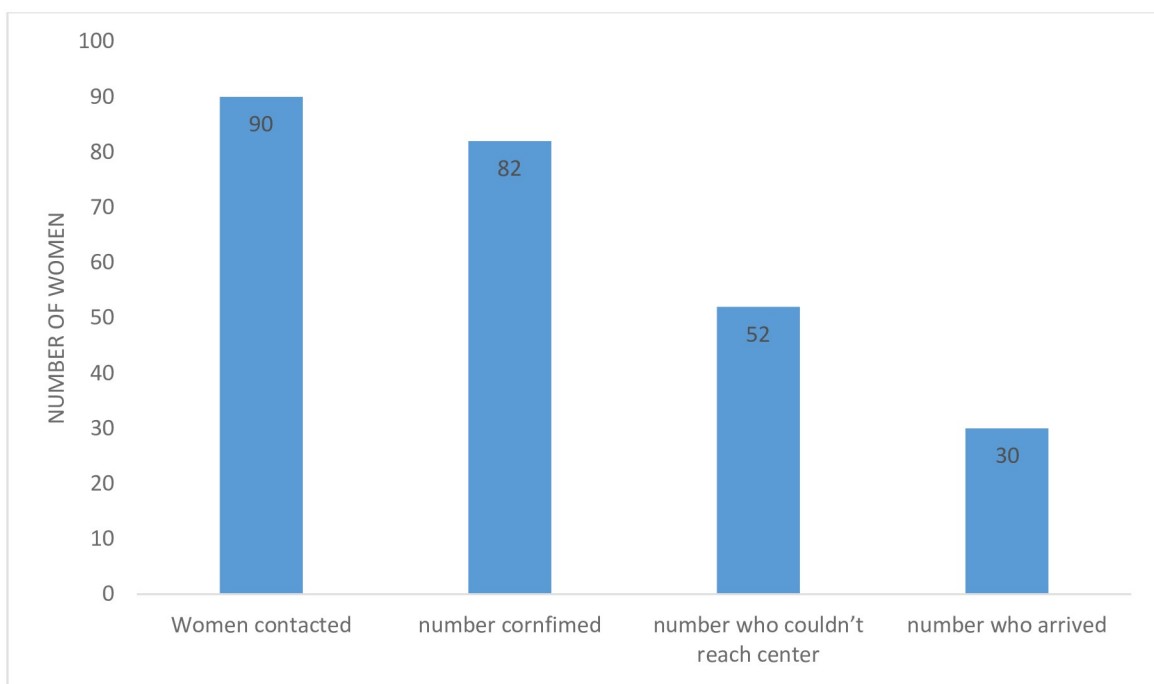

**Fig 2. Cascade of the process from being called to attend the camp to actual attendance.**

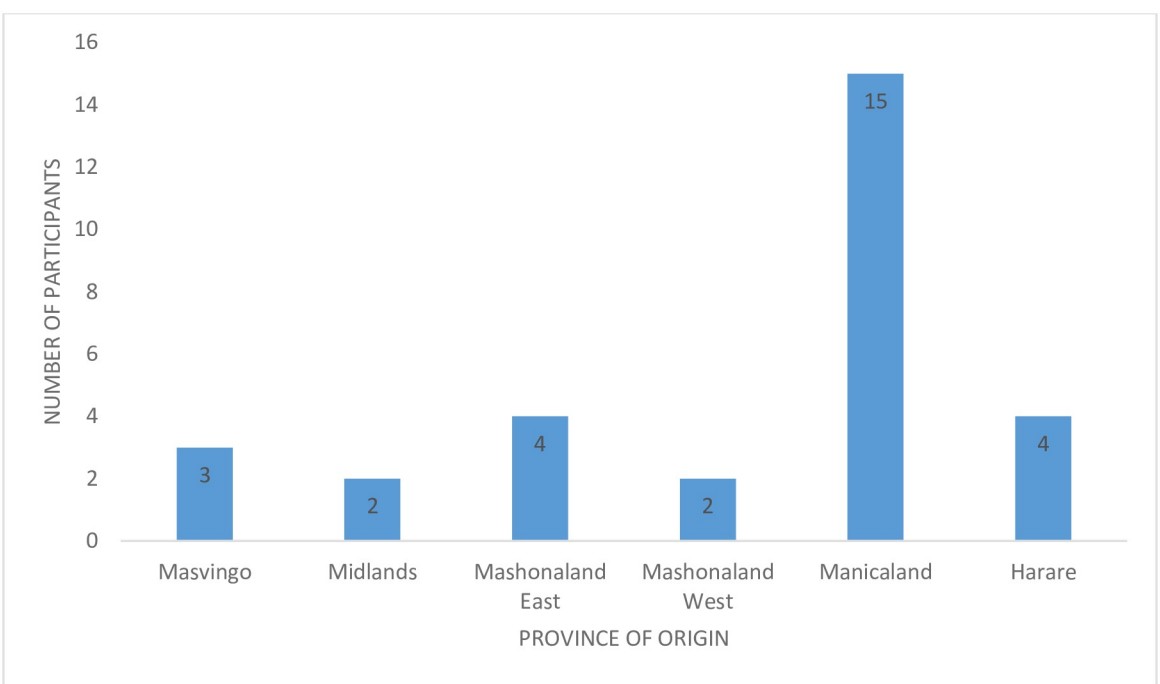

**Fig 3. Distribution of participants by province of origin.**

- Afraid of law enforcement agents—15
- Were turned back by law enforcement agents at check-points—13

This left only 30 women who actually attended the camp. This is against the initial plan of repairing 90 women. Fig 3 below shows the distribution of fistula camp attendees by province. Fig 3 shows that half (15 of the 30) of study participants came from Manicaland province.

## Likelihood of attending the camp by province of origin

At 95% confidence level, participants from Manicaland and Masvingo provinces were about 11 times more likely to reach the repair center than those coming from other provinces (OR 11.5; CI 3.7–35.3) and this difference was statistically significant (p<0.001), (Masvingo and Manicaland were put together because although Mashoko Hospital is in Masvingo, it is at the far border separating the two provinces).

## Likelihood of attending the camp by age group

Adolescent and young participants (below age 25 years) were 0.3 times more likely to attend the camp than those 25years and older (OR1.3; CI 0.53–3.32) but this was not statistically significant (p = 0.54).

Similar results were observed for period living with fistula and the likelihood of attending the camp where the OR was 1.06, CI 0.48–4.7, p = 0.8.

On arrival, all women were screened for Covid-19 infection symptomatically. They had temperature checks and asked questions on whether they had symptoms. They had a rapid test for the infection. Three of the thirty women got a positive rapid test result.

They were immediately isolated and specimens for DNA PCR tests were collected and sent for testing. These came out negative and so the women were removed from isolation.

**Table 1. Demographic characteristics of participants.**

| Age groups | | |
|---|---|---|
| 15–19 | 20–24 | 25+ |
| 6 | 7 | 17 |
| **Time staying with incontinence** | | |
| <1year | 1–5 years | >5 years |
| 11 | 8 | 11 |
| **Type of referral** | | |
| Health facility | Community-based workers | Radio |
| 4 | 22 | 4 |

Measures for infection prevention and control were strictly observed, i.e. frequent hand-washing and sanitization, social distancing and wearing of facemasks. The women were admitted in batches of five per day and a ward which used to accommodate 30 patients was able to accommodate only 10.

Other demographic data are presented in Table 1 below.

Table 1 shows that adolescents and young women below the age of 25 contributed a significant proportion (43%) of affected women with 6 participants being below 20 years. The oldest was 73 years old and the youngest was 17. The majority (19 women) of the participants developed the incontinence problem in the 5 years preceding this study.

The most recent fistula had a duration of 3 months while the longest duration was 39 years. Most (22 women) of the study participants were referred by community based workers.

Of the 30 women who attended the fistula repair camp, five had fecal incontinence due to 3rd and 4th degree perineal tears and 25 had fistula.

Table 2 above shows that the majority (20 women) of the participants suffered urinary incontinence. Of the 30 study participants, five had incontinence of stool due to 3rd and 4th degree perineal tears so 25 were actual fistula cases.

Only eight participants had tried to get treatment of their fistula before.

## Surgical outcomes among the 25 clients repaired at Mashoko hospital

The outcome of surgery upon discharge was classified into three categories

1. Dry—fistula was successfully closed and there is no leaking

2. Incontinence—surgery was successful in closing the fistula hole but patient is discharged leaking due to stress incontinence

3. Fistula not closed–fistula was not closed (a dye test is done to distinguish leaking due to incontinence or unsuccessful surgery).

**Table 2. Type of incontinence and type of injury.**

| Type of incontinence | | | | | |
|---|---|---|---|---|---|
| Fecal | Urinary | | | | |
| 10 | 20 | | | | |
| **Type of injury** | | | | | |
| 3rd and 4th degree perineal tears | Vesico vaginal fistula | Circumferential fistula | Urethra-vaginal fistula | Rectovaginal fistula | Vesico-cervical fistula |
| 5 | 13 | 6 | 1 | 4 | 1 |

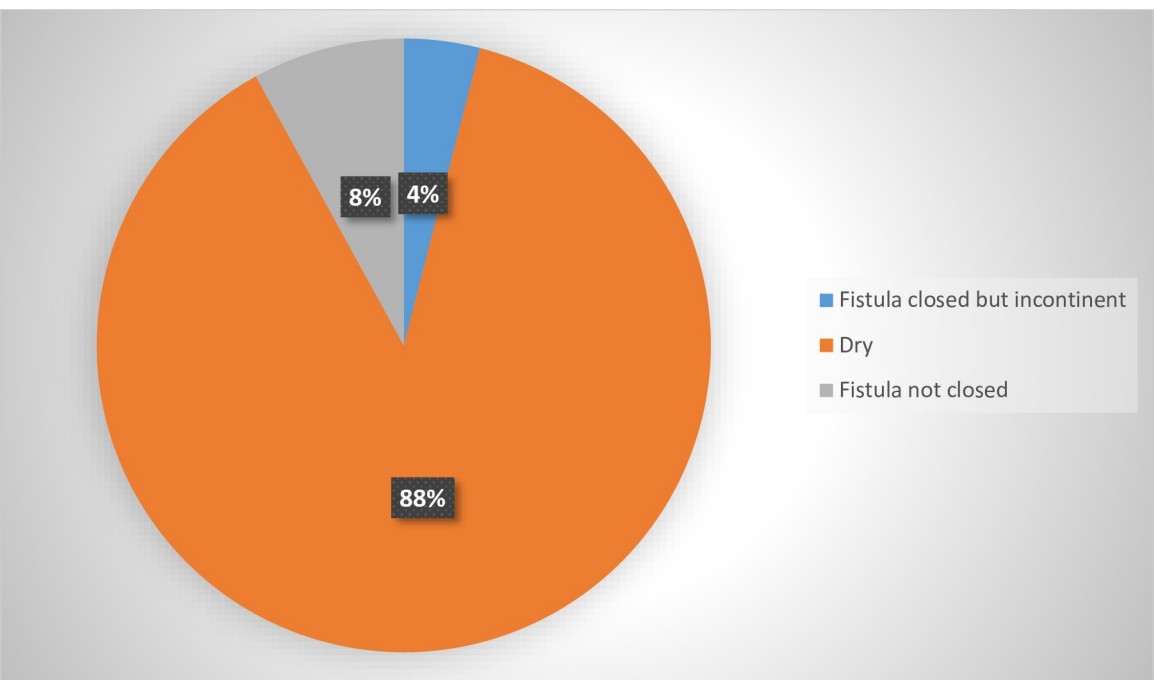

**Fig 4. Treatment outcomes for fistula cases (n = 25).**

Those with perineal tears were repaired accordingly.

Fig 4 below shows the surgical treatment outcomes for fistula cases.

Fig 4 shows that 22 women, which were the majority of the participants, went home dry. The fistula were closed and they no longer experienced incontinence, two participants had failed surgery, their fistula were not closed. The treatment success rate was 92%.

All participants were discharged from hospital without symptoms of Covid-19. No further diagnostic tests were done for Covid-19.

## Discussion

The main fistula repair center in Zimbabwe, Chinhoyi Provincial Hospital stopped conducting fistula repair surgeries because of the Covid-19 pandemic. This was necessitated by the need to control the pandemic. Infection prevention and control measures which included social distancing, frequent hand sanitization and wearing of face masks were strictly observed. These measures meant cutting on most non-emergency surgeries so as to avoid unnecessary human contact and creating space for anticipated rise in Covid-19 case load. This was a trend noted in other hospitals. A study conducted in October 2020 at a tertiary UK hospital found that the hospital reduced non-emergency operations by 44% due to Covid-19 [5]. A study on the effect of Covid-19 pandemic on health services in Nigeria also found out that the pandemic had thrown health systems into disarray and most services which were deemed not life-saving were put on hold and most efforts directed to the fight of the pandemic [6].

The attendance rate of women to the repair camp was greatly affected by the Covid-19 pandemic. Most of the women (43%) came from Manicaland Province alone. This may be attributed to the increased efforts in active surveillance in the province. The introduction and scaling up of the community fistula surveillance program has increased the awareness of fistula as a treatable condition in Manicaland. Knowledge of obstetric fistula as a medical condition

which can be treated and awareness of availability of treatment services has been shown to increase uptake of fistula repair services [7,8]. In a report of a two-week fistula campaign in Nigeria, the highly publicized event saw more than 500 fistula survivors being treated [9].

The fact that most fistula cases came from Manicaland may not necessarily mean that the province has the highest fistula burden in the country but awareness raising, active surveillance and publicizing of fistula repair program in the province may help explain this observation.

Active search for fistula cases was being done by community based workers, fistula survivors and local health workers and this may have helped the community to receive the information and respond better. This phenomenon was also noted in a study documenting the lived experiences of women with fistula in Kenya [10]. The researchers found that women opened up better to locally based health workers and local community workers.

As noted in other studies, a significant proportion of participants were adolescent and young women aged 24 years and below [11]. In a study conducted in Kaptembwa-Nakuru in Kenya it was noted that only 2.04% of the responds were above 25 years of age [11].

In their study documenting the experiences of women living with obstetric fistula in 2008, Semere and Nour argued that the body of a girl below the age 20 years is not mature enough to withstand the burden of child bearing, therefore they are at increased risk of developing fistula [12]. Contrary to this finding, a study on fistula survivors' experiences in Ghana had only one for the ten participants below 25 years [8]. These differences could be due to sampling issues. However, the World Health Organization (2019) and the United Nations Population Fund (2015) [1,4] recognize childbirth during tender ages below age 25 years as high risk for developing obstetric fistula [1,4].

The findings of this study revealed that most participants had developed fistula 5 years or less prior to data collection for this study, with the majority of the participants having developed the problem less than 1 year ago. This finding is of concern as it shows that obstetric fistula continues to be a significant complication of pregnancy in Zimbabwe. The country has one district hospital (comprehensive emergency obstetric and neonatal care facility) for every 250,000 population (14). This is twice the recommended level for such facilities of 1 per 500,000 population (15). In theory, Zimbabwe should have very few or no obstetric fistula cases. This observation has a reference to the strength and effectiveness of the national health system as noted by other studies and the WHO (2014) that the continued occurrence of obstetric fistula is an indication of weak health systems [13–15].

Although there were other means of participants' referral to the fistula center, the majority (20 out of 28) were referred by community based workers in Manicaland through the community surveillance system. The majority of the women identified and referred through this system actually had fistula i.e. 19 of the 20. Only one had stool incontinence due to a 4th degree perineal tear. Although it may be too soon and the numbers inadequate to evaluate the specificity of the community fistula screening tool, these results show promising specificity threshhold of the tool.

This study noted that most of the participants had vesico-vaginal fistula and most suffered urinary incontinence, this is in agreement with findings of other studies and documented literature [16]. The high treatment success rate observed by this study, 92%, gives hope to fistula survivors for restoration of good health and dignity after repair.

Other studies also noted that when surgical repair of fistula is performed by experienced surgeons, the treatment success rates are very high [17,18] This observation is believed to motivate fistula survivors to take up e treatment services especially if more survivors who had successful repair advocate for treatment.

Proximity to the fistula repair center was the main determinant for whether the participants would reach the center or not. This may be so due to the number of law enforcement agents

checkpoints participants had to pass on their way to hospital. Distance to health facilities which offer services was also noted by the World Health Organization to affect uptake of services [15]. This was made more evident by the Covid-19 pandemic lockdown restrictions.

## Conclusions

With the above findings, this study concluded that the Covid-19 pandemic negatively affected the fistula repair program in the country as it was considered one of the non-essential services. Women living with obstetric fistula are there in communities as shown by the Manicaland case, they need to be identified, given information on available treatment services and linked to fistula care programs for repair, even during the Covid-19 pandemic.

It also concluded that community based workers and fistula survivors can be instrumental in community surveillance of fistula cases in their communities when they are trained on how to use standard surveillance tools.

## Recommendations to programmers and policy makers

Basing on the findings of this study, it is recommended to the Ministry of Health and Child Care to scale up and intensify fistula case finding in communities and link them to treatment services so as to end or minimize suffering of women and girls. There is need to increase treatment centers so that patients will not need to travel long distances to access services, this need has been highlighted by the Covid-19 lockdown. It is also recommended to scale up the community obstetric fistula surveillance system making use of community based workers and other personnel to actively search for fistula cases in the community. The use of mobile communications through phone calls or social network platforms is recommended to sustain screening and identification of women living with fistula so as to have them repaired as soon as possible after the Covid-19 pandemic or when the situation allows. The program needs to be expanded to establish fistula repair centers in other provinces for equity and accessibility even in times like the Covid-19 pandemic lockdown.

## Study limitations

Data collection was mainly records review and telephone conversations, this approach poses restrictions on collecting the finer details on information obtained as compared to face-to-face interviews. Also, a larger sample size would have provided more confidence in obtained results.

## Supporting information

**S1 File. Key informant indepth interview guide.**
(DOCX)

**S2 File. Q4 2020 fistula log.**
(XLSX)

## Acknowledgments

The authors acknowledge contributions of Sandra Murwira for assisting in conducting data collection from the key informants.

## Author Contributions

**Conceptualization:** Chipo Chimamise, Stephen Peter Munjanja, Mazvita Machinga, Iris Shiripinda.

**Data curation:** Chipo Chimamise, Stephen Peter Munjanja.

**Formal analysis:** Chipo Chimamise, Stephen Peter Munjanja, Mazvita Machinga, Iris Shiripinda.

**Methodology:** Chipo Chimamise, Stephen Peter Munjanja, Mazvita Machinga, Iris Shiripinda.

**Project administration:** Chipo Chimamise.

**Resources:** Chipo Chimamise.

**Supervision:** Stephen Peter Munjanja, Mazvita Machinga, Iris Shiripinda.

**Visualization:** Chipo Chimamise.

**Writing – original draft:** Chipo Chimamise.

**Writing – review & editing:** Stephen Peter Munjanja, Mazvita Machinga, Iris Shiripinda.

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
