## [Decision Letter · Decision Letter 0]

11 Mar 2021

PONE-D-21-05832

Impact of Covid-19 pandemic on obstetric fistula repair programming and characteristics of Obstetric fistula repair camp attendees at Mashoko Christian Hospital in Zimbabwe

PLOS ONE

Dear Chipo

Thank you for submitting your manuscript to PLOS ONE. After careful consideration, we feel that it has merit but does not fully meet PLOS ONE’s publication criteria as it currently stands. Therefore, we invite you to submit a revised version of the manuscript that addresses the points raised during the review process.

We look forward to receiving your revised manuscript.

Kind regards,

Godfrey Nyangadzayi Musuka

Academic Editor

PLOS ONE

Journal Requirements:

Additional Editor Comments:

1.Please address all comments from reviewers

2. Please beef up the section on weaknesses of the study

3. Please include a section on recommendations to programmers and policymakers

Reviewers' comments:

Reviewer's Responses to Questions

**Comments to the Author**

1. Is the manuscript technically sound, and do the data support the conclusions?

Reviewer #1: Yes

Reviewer #2: Yes

2. Has the statistical analysis been performed appropriately and rigorously? 

Reviewer #1: Yes

Reviewer #2: No

3. Have the authors made all data underlying the findings in their manuscript fully available?

Reviewer #1: Yes

Reviewer #2: Yes

4. Is the manuscript presented in an intelligible fashion and written in standard English?

Reviewer #1: Yes

Reviewer #2: Yes

5. Review Comments to the Author

Reviewer #1: This is a very good manuscript that attempts to describe the impact of the COVID-19 pandemic on obstetric fistula repair in Zimbabwe. Obstetric fistulae remain a significant challenge in Zimbabwe and attempts to repair them are commendable. This is a technically sound, original manuscript, which has a significant contribution to the literature regarding the impacts of the COVID-19 pandemic on public health services delivery in a resource-limited setting. This study is worthy publishing; however, the author must correct the many grammatical errors noted in the manuscript, which are highlighted on the attachment.

Reviewer #2: The manuscript is a relevant piece of academic writing, however the statistical analysis is too simplistic as the authors chose to limit themselves to univariate analysis. Beyond the reasons provided by the clients one might want to know whether there were any factors associated with a no show for fistulae repair(only 1/3 came)? Is it possible that clients with certain characteristics were more likely not to show up for repair than others? These questions would give insights into issues of equity which are relevant for such a program undertaking. Such questions would have been answered through multivariate analysis and a sample size of 90 must be powered enough to give insight. Given that a complete dataset is available I suggest that the authors consider adding a more detailed statistical analysis beyond the descriptives they shared

6. PLOS authors have the option to publish the peer review history of their article (what does this mean?). If published, this will include your full peer review and any attached files.

Reviewer #1: **Yes: **Grant Murewanhema

Reviewer #2: **Yes: **Brian Kumbirai Moyo

---

## [Author Response · Author response to Decision Letter 0]

16 Mar 2021

Reference list reviewed, the reference which was not complete was corrected ie ref 13. Abokaiagana A. Experiences of women with Obstetric fistula in the Bawku East District of the Upper East Region, Ghana [Internet]. University of Ghana; 2010 [cited 2021 Jan 18]. Available from: http://ugspace.ug.edu.gh. 

It was originally cited as: 8. Abokaiagana A. Experiences of women with Obstetric fistula in the Bawku East District of the Upper East Region. :155. 

Figure file naming was corrected. Fig 1was renamed Fig 1.tif, Fig 2 was renamed Fig 2.tif, Fig 3 was renamed Fig 3.tf and Figure 4 was renamed Fig 4.tif

• Please address all comments from reviewers

This was done

• Please beef up the section on weaknesses of the study

This was attended to: 

Study limitations

Data collection was mainly records review and telephone conversations, this approach poses restrictions on collecting the finer details on information obtained as compared to face-to-face interviews. Also, a larger sample size would have provided more confidence in obtained results.

• Please include a section on recommendations to programmers and policymakers

This was addressed:

Recommendations to programmers and policy makers

Basing on the findings of this study, it is recommended to the Ministry of Health and Child Care to scale up and intensify fistula case finding in communities and link them to treatment services so as to end or minimize suffering of women and girls. There is need to increase treatment centers so that patients will not need to travel long distances to access services, this need has been highlighted by the Covid-19 lockdown. It is also recommended to scale up the community obstetric fistula surveillance system making use of community based workers and other personnel to actively search for fistula cases in the community. The use of mobile communications through phone calls or social network platforms is recommended to sustain screening and identification of women living with fistula so as to have them repaired as soon as possible after the Covid-19 pandemic or when the situation allows. The program needs to be expanded to establish fistula repair centers in other provinces for equity and accessibility even in times like the Covid-19 pandemic lockdown.

• Reviewer #1: This is a very good manuscript that attempts to describe the impact of the COVID-19 pandemic on obstetric fistula repair in Zimbabwe. Obstetric fistulae remain a significant challenge in Zimbabwe and attempts to repair them are commendable. This is a technically sound, original manuscript, which has a significant contribution to the literature regarding the impacts of the COVID-19 pandemic on public health services delivery in a resource-limited setting. This study is worthy publishing; however, the author must correct the many grammatical errors noted in the manuscript, which are highlighted on the attachment.

This was attended to as highlighted in the copy with track changes

• Reviewer #2: The manuscript is a relevant piece of academic writing, however the statistical analysis is too simplistic as the authors chose to limit themselves to univariate analysis. Beyond the reasons provided by the clients one might want to know whether there were any factors associated with a no show for fistulae repair(only 1/3 came)? Is it possible that clients with certain characteristics were more likely not to show up for repair than others? These questions would give insights into issues of equity which are relevant for such a program undertaking. Such questions would have been answered through multivariate analysis and a sample size of 90 must be powered enough to give insight. Given that a complete dataset is available I suggest that the authors consider adding a more detailed statistical analysis beyond the descriptives they shared

Attended to. Authors went further to perform bivariate analysis (only one factor was statistically significant) and discussed the findings in the discussion section and cascaded it to recommendations.

In results:

Likelihood of attending the camp by province of origin

At 95% confidence level, participants from Manicaland and Masvingo provinces were about 11 times more likely to reach the repair center than those coming from other provinces (OR 11.5; CI 3.7-35.3) and this difference was statistically significant (p<0.001), (Masvingo and Manicaland were put together because although Mashoko Hospital is in Masvingo, it is at the far border separating the two provinces). 

Likelihood of attending the camp by age group

Adolescent and young participants (below age 25 years) were 0.3 times more likely to attend the camp than those 25years and older (OR1.3; CI 0.53-3.32) but this was not statistically significant (p=0.54).

Similar results were observed for period living with fistula and the likelihood of attending the camp where the OR was 1.06, CI 0.48-4.7, p = 0.8.

In discussion:

Proximity to the fistula repair center was the main determinant for whether the participants would reach the center or not. This may be so due to the number of law enforcement agents checkpoints participants had to pass on their way to hospital. Distance to health facilities which offer services was also noted by the World Health Organization to affect uptake of services [15]. This was made more evident by the Covid-19 pandemic lockdown restrictions.

In recommendations:

The program needs to be expanded to establish fistula repair centers in other provinces for equity and accessibility even in times like the Covid-19 pandemic lockdown.

---

## [Editor Report · Decision Letter 1]

18 Mar 2021

Impact of Covid-19 pandemic on obstetric fistula repair program in Zimbabwe

PONE-D-21-05832R1

Dear Dr. CHIMAMISE,

We’re pleased to inform you that your manuscript has been judged scientifically suitable for publication and will be formally accepted for publication once it meets all outstanding technical requirements.

Kind regards,

Godfrey Nyangadzayi Musuka

Academic Editor

PLOS ONE
---

## [Editor Report · Acceptance letter]

23 Mar 2021

PONE-D-21-05832R1 

Impact of Covid-19 pandemic on obstetric fistula repair program in Zimbabwe 

Dear Dr. Chimamise:

I'm pleased to inform you that your manuscript has been deemed suitable for publication in PLOS ONE. Congratulations! Your manuscript is now with our production department. 

Kind regards, 

on behalf of

Dr. Godfrey Nyangadzayi Musuka 

Academic Editor

PLOS ONE